# Review of Treatment Options for the Management of Advanced Stage Hodgkin Lymphoma

**DOI:** 10.3390/cancers13153745

**Published:** 2021-07-26

**Authors:** Hélène Vellemans, Marc P. E. André

**Affiliations:** Department of Hematology, CHU UCL, 530 Yvoir, Belgium; marc.andre@uclouvain.be

**Keywords:** Hodgkin lymphoma, ABVD, BEACOPP, [^18^F]FDG-PET/CT

## Abstract

**Simple Summary:**

The cure rate of Hodgkin lymphoma is currently higher than 80% for almost all stages at diagnosis. Despite the particularly good efficacy of chemotherapy and radiotherapy, some late complications such as cardiovascular disease and second malignancies can occur in a small proportion of patients. A major concern nowadays is, therefore, to find the balance between remission and toxicity in the development of new treatments for classical Hodgkin lymphoma. This review focuses on how to best treat first-line advanced Hodgkin lymphomas, considering the acute and long-term consequences of chemotherapy and radiotherapy treatments. New drugs such as brentuximab vedotin and checkpoint inhibitors are also a field of interest.

**Abstract:**

Hodgkin lymphoma (HL) is a lymphoid-type hematologic disease that is derived from B cells. The incidence of this lymphoid malignancy is around 2–3/100,000/year in the western world. Long-term remission rates are linked to a risk-adapted approach, which allows remission rates higher than 80%. The first-line treatment for advanced stage classical HL (cHL) widely used today is doxorubicin, bleomycin, vinblastine, and dacarbazine (ABVD) or escalated bleomycin, etoposide, doxorubicin, cyclophosphamide, vincristine, procarbazine, and prednisone (BEACOPP_esc_) chemotherapy. Randomized studies comparing these two regimens and a recently performed meta-analysis have demonstrated consistently better disease control with BEACOPP_esc_. However, this treatment is not the standard of care, as there is an excess of acute hematological toxicities and therapy-related myeloid neoplasms. Moreover, there is a recurrent controversy concerning the impact on overall survival with this regimen. More recently, new drugs such as brentuximab vedotin and checkpoint inhibitors have become available and have been evaluated in combination with doxorubicin, vinblastine, and dacarbazine (AVD) for the first-line treatment of patients with advanced cHL with the objective of tumor control improvement. There are still major debates with respect to first-line treatment of advanced cHL. The use of positron emission tomography-adapted strategies has allowed a reduction in the toxicity of chemotherapy regimens. Incorporation of new drugs into the treatment algorithms requires confirmation.

## 1. Introduction

Hodgkin lymphoma (HL) is a B-cell lymphoma, accounting for about 10–15% of newly diagnosed lymphoma cases in the United States (8260 of 80,500 cases), and the remainder are non-Hodgkin lymphoma [1]. The incidence is 2–3/100,000/year in the western world. HL mainly affects young people between the ages of 20 and 30 years but can also affect more elderly people. Men are affected slightly more frequently than women. There are two types of HL: classical Hodgkin lymphoma (cHL) and nodular lymphocyte-predominant Hodgkin lymphoma. Classical HL accounts for approximately 95% of all HLs and is subdivided into four histologic subtypes: nodular sclerosis, mixed cellularity, lymphocyte-rich, and lymphocyte-depleted [2]. The Ann Arbor system is most commonly used for HL staging. The different staging criteria are lymph node involvement on one or both sides of the diaphragm, the presence of contiguous extranodal involvement or extranodal disease, presence of B symptoms, and existence of a bulky disease. In Europe, the definition of advanced stage HL varies by research group and is slightly different for the Lymphoma Study Association (LYSA)/European Organization for Research and Treatment of Cancer (EORTC) and the German Hodgkin Study Group (GHSG) because risk factors are defined differently [3] (Table 1).

The treatment of HL has evolved considerably over the past 40 years. Involved-field and involved-node radiotherapy, multi-agent chemotherapy, and combined chemoradiotherapy have improved patient care. Recent approaches such as risk-adapted and response-adapted modulation have allowed more personalized treatment. The introduction of antibody–drug conjugates and immune checkpoint inhibitors is also a considerable step forward [4]. In actuality, the risk-adapted approach allows remission rates ≥ 80% for almost all stages at diagnosis [5]. In recent trials, and probably in real life, it is known that patients ≤ 50 years of age mostly die for reasons other than the disease itself; however, those ≥60 years of age still die more frequently because of the disease [4]. Thus, despite the particularly good efficacy of radiotherapy and chemotherapy, some late complications such as cardiovascular disease and second malignancies can occur in a small proportion of patients [6]. In this article, we focus on the management of advanced stage cHL in patients aged between 16 and 60 years, with special attention to the balance between cure and toxicity. Indeed, it remains one of the main issues in the development of improved treatment strategies for cHL patients.

## 2. ABVD versus BEACOPP

Advanced stage cHL is usually treated with chemotherapy alone—most commonly, either with escalated bleomycin, etoposide, doxorubicin, cyclophosphamide, vincristine, procarbazine, and prednisone (BEACOPP_esc_) or doxorubicin, bleomycin, vinblastine, and dacarbazine (ABVD) chemotherapy. Repeat cycles are administered every 3 weeks for BEACOPPesc and every 2 weeks for ABVD. The term “escalated” means that the doses of treatment are higher than those of standard BEACOPP (BEACOPP_std_). Additional radiotherapy is limited to patients with residual disease after chemotherapy.

Many randomized studies and a recent meta-analysis comparing BEACOPP_esc_ and ABVD regimens have demonstrated consistently better disease control with BEACOPP_esc_. These studies are summarized as follows (Table 2).

In the EORTC 20012 Intergroup Trial, Carde and colleagues [7] compared eight cycles of ABVD versus four cycles of BEACOPP_esc_ plus four cycles of BEACOPP_std_ (BEACOPP_4+4_) without radiotherapy in patients aged 16–60 years with stage III or IV cHL. Event-free survival and overall survival (OS) were similar with both regimens. Hematological toxicity was more significant with BEACOPP_4+4_ compared to ABVD, with more frequent febrile neutropenia (6.3% vs. 0%, respectively). The cumulative incidences of late side effects and second malignancies were not significantly different.

The HD2000 Trial [8,9] compared three regimens in patients with advanced HL: six cycles of ABVD, four cycles of BEACOPP_esc_ plus two cycles of BEACOPP_std_, and six cycles of cyclophosphamide, lomustine, vindesine, melphalan, prednisone, epidoxorubicin, vincristine, procarbazine, vinblastine, and bleomycin (COPP-EBV-CAD (CEC)). The BEACOPP regimen resulted in superior 10-year progression-free survival (PFS) than ABVD did, but with higher rates of grade III–IV neutropenia (54% vs. 34%; *p* = 0.016) and severe infections (14% vs. 2%; *p* = 0.003). Long-term follow-up showed 13 second malignancies, which were distributed as follows: ABVD (*n* = 1), BEACOPP (*n* = 6), and CEC (*n* = 6). The high rate of second malignancies with BEACOPP confirms the late toxicity with this regimen. All of the second malignancies except one were diagnosed in patients in complete remission (CR), and unfortunately, eight resulted in death. Five of the 13 patients had received radiotherapy as part of their initial treatment. A solid tumor close to an irradiated field was diagnosed in three of them. The median time from the end of HL treatment to diagnosis of second malignancy was 90 months (range, 4–153 months).

In the LYSA H34 randomized trial, Mounier and colleagues [10] compared eight cycles of ABVD and four cycles of BEACOPP_esc_ plus ≥ 4 cycles of BEACOPP_std_. The 5-year event-free survival rate was 77% for the BEACOPP regimen and 62% for ABVD (hazard ratio (HR) = 0.6, *p* = 0.07). Five-year PFS was significantly higher with the BEACOPP regimen (93% versus 75%; HR = 0.3, *p* = 0.007). OS at 5 years was 99% with the BEACOPP regimen versus 92% with ABVD (HR = 0.18, *p* = 0.06). Treatment with the BEACOPP regimen resulted in increased immediate morbidity, with a higher rate of febrile neutropenia compared to ABVD (35% vs. 8%, respectively).

Viviani and colleagues [11] compared six or eight cycles of ABVD and eight cycles of BEACOPP_4+4_. A total of 331 patients with unfavorable HL were enrolled. Initial tumor control was better with BEACOPP_4+4_. The 7-year OS rate was not significantly different when comparing the groups (89% in the BEACOPP_4+4_ arm and 85% in the ABVD arm, *p* = 0.39).

In a meta-analysis of the four previously described trials, André and colleagues [12] confirmed that BEACOPP offers better initial disease control. The 7-year PFS rate was 81.1% for BEACOPP and 71.1% for ABVD. Two time periods were defined for the analysis of OS as the HR was not constant over time. In the first time period up to 18 months, there was no clear difference in OS between ABVD and BEACOPP. For the patients surviving 18 months, there was a slight OS difference in favor of BEACOPP. In the ABVD group, three of five deaths were due to HL. In the BEACOPP arm, the main cause of death was a second cancer, such as myelodysplastic syndrome or acute myeloid leukemia (representing approximately one-third of the deaths).

In a systematic review and meta-analysis, Skoetz and colleagues [13] demonstrated that OS was significantly better with six cycles of BEACOPP_esc_ than with ABVD and other regimens. They searched the Cochrane library, Medline, and conference proceedings between January 1980 and June 2013. They found 14 eligible trials that assessed 11 different regimens in approximately 10,000 patients. OS was superior with six cycles of BEACOPP_esc_ than with the other regimens (HR = 0.38, 95% confidence interval (CI) 0.20–0.75) with a survival benefit of 7% for 5-year OS [13].

Although BEACOPP appears to be better for disease control, it is not a standard of care because of the increased acute hematological toxicity and therapy-related myeloid neoplasms. Moreover, there is some degree of controversy around the impact of this regimen on OS [12,13]. Based on the results of the aforementioned studies, BEACOPP and ABVD have advantages and disadvantages. For the two regimens, altered reproductive function in young women and men is also a key concern. In the literature, hormonal levels (high level of follicle-stimulating hormone and/or low level of anti-Müllerian hormone) reflecting reduced ovarian reserve and amenorrhea indicate impaired fertility in the majority of women treated with BEACOPP. After this regimen, the recovery time of spermatogenesis in men was also much longer. Young patients treated with BEACOPP should be informed of its high gonadotoxicity and directed to a specialist to discuss protective methods to preserve fertility [14,15,16].

Finally, in the case of relapse after ABVD, salvage therapy has consequent severe early and late toxic effects that are not dissimilar to those with BEACOPP. The patient must therefore be warned of the choice between these different regimens and the consequences that result from them [11]. With these considerations, the goal of the following trials for BEACOPP_esc_ was to reduce acute and late toxicities while maintaining the high rate of tumor control. The field of advanced cHL treatment has moved to a risk-adapted approach. For ABVD, the goals of the next generation of trials are to reduce toxicity but also to improve efficacy.

## 3. PET-Adapted Treatment

^18^-Fluoro-deoxyglucose positron emission tomography combined with computed tomography ([^18^F]FDG-PET/CT) is now widely used in the management of HL patients. It is useful to determine the stage of the disease at baseline and to evaluate early chemosensitivity to first-line treatment. Compared to conventional CT, sensitivity with [^18^F]FDG-PET/CT is increased by 10–20%, resulting in an upstaging rate of 10–40% and a modified treatment decision in up to 20% of patients [17].

In a retrospective series of patients receiving ABVD studied by Hutchings and colleagues [18], those with a metabolic remission appeared to have a better prognosis than those with persistently FDG-avid disease after two cycles of treatment, supporting interim [^18^F]FDG-PET/CT as a means of predicting outcome for HL. Early interim FDG-PET results after two cycles (PET-2) were prognostically stronger than previously established prognostic factors. Early response on [^18^F]FDG-PET/CT was predictive of both primary treatment response and survival, with a 2-year PFS for PET2-negative patients of 96.0% compared to 0% for PET-positive patients.

Gallamini and colleagues [19] enrolled 108 patients in a study of newly diagnosed HL at 11 Italian hematology institutions. Disease stages were IIA with adverse prognostic factors or stages IIB–IV. The goal was to explore the predictive value on therapy outcome with a [^18^F]FDG-PET/CT performed early after two cycles of ABVD. Positive PET-2 and negative PET-2 were seen in 20 and 88 patients, respectively. Of the patients with negative PET-2, 97% remained in CR. The remaining 3% progressed or relapsed rapidly after the end of the chemotherapy. In this situation, the positive predictive value of PET-2 was 90% and the negative predictive value was 97%. The 2-year probability of failure-free survival was 96% for PET-2-negative and 6% for PET-2-positive patients (log-rank test = 116.7, *p* < 0.01).

For interpretation criteria, there are two scales of interest. The first is the Deauville five-point scale, which is an internationally recommended scale that is now widely documented in the literature [20]. In June 2011, a workshop was held at the International Conference on Malignant Lymphoma (ICML) in Lugano. The purpose was, among others, to update the recommendations for the evaluation, staging, and response assessment of patient with HL. A newer classification was established based on a five-point scale called the “Lugano Criteria” [21]. It determines a complete metabolic response (CMR) for scale scores 1–3 and no metabolic response (no change from baseline) and a partial response (reduced uptake compared to baseline) or progressive disease (increased uptake compared to baseline or new lesion) for scale scores 4 and 5. This Lugano classification has shown good prognostic discrimination and good inter-observer reproducibility. The classification can thus be used to standardize [^18^F]FDG-PET/CT reports [22,23]. In the literature, there are two types of strategies after an early [^18^F]FDG-PET/CT. The first is the de-escalation strategy, which evaluates treatment de-escalation for good metabolic responders after two cycles of BEACOPP_esc_. The second approach, called an escalation strategy, proposes strengthening treatment for poor responders after two cycles of ABVD by introducing the BEACOPP_esc_ regimen [23] (Figure 1).

In the HD18 Trial [24], the GHSG investigated whether the metabolic response determined by PET-2 after two cycles of BEACOPP_esc_ would allow further adaptation of treatment intensity, increasing it for PET-2-positive (score 3–5) patients and reducing it for PET-2-negative (score 1–2) patients. The 5-year PFS for PET-2-negative patients was 90.8% (95% CI 87.9–93.7) for 8/6 × BEACOPP_esc_ and 92.2% (95% CI 89.4–95.0) for 4 × BEACOPP_esc_ (difference 1.4%, 95% CI −2.7 to 5.4). The results showed fewer severe infections (40/498 (8%) vs. 75/502 (15%)) and organ toxicities (38/498 (8%) vs. 91/502 (18%)) in patients receiving 4 × BEACOPP_esc_. Thus, for patients with negative PET-2, the reduction to four cycles of BEACOPP_esc_ provided outstanding efficacy and increased OS by reducing treatment-related risks.

In the previously described trial, patients with a persistent mass after chemotherapy measuring ≥ 2.5 cm and positive on [^18^F]FDG-PET/CT received additional radiotherapy. Nevertheless, a positive [^18^F]FDG-PET/CT after effective chemotherapy was associated with higher risk of subsequent treatment failure, even though PET-positive patients were treated with additional radiotherapy and PET-negative patients were not [24].

Next, the LYSA Group published the AHL2011 Trial, which investigated whether [^18^F]FDG-PET/CT monitoring during treatment could allow dose de-escalation by switching the regimen (BEACOPP_esc_ to ABVD) in early responders without loss of disease control compared to standard treatment without [^18^F]FDG-PET/CT monitoring [25]. On intention-to-treat analysis, the estimated 5-year PFS was similar in the standard treatment group (86.2%, 95% CI 81.6–89.8%) and the PET-driven treatment group (85.7%, 81.4–89.1%) (stratified HR = 1.084, 95% CI 0.737–1.596, *p*_non-inferiority_ = 0.65; unstratified HR = 1.066, 95% CI 0.725–1.569, *p*_non-inferiority_ = 0.63). In conclusion, [^18^F]FDG-PET/CT used after two cycles of BEACOPP_esc_ allows the use of a response-adapted strategy with the possibility of safely delivering four cycles of ABVD without impairing disease control in patients who achieve early response to treatment. As described previously, patients with a Deauville score 1 and 2 in the HD18 Trial were considered as PET-2-negative and those with a Deauville score 3 as PET-2-positive. In this context, only 51% of patients had negative PET-2 compared with 87% in the AHL2011 Trial.

Groups using ABVD as the initial therapy have also examined the possibility of reducing toxicity, especially in patients with a negative interim [^18^F]FDG-PET/CT. In the RATHL Trial [26], patients with newly diagnosed advanced cHL underwent a baseline [^18^F]FDG-PET/CT, received two cycles of ABVD chemotherapy, and then underwent an interim [^18^F]FDG-PET/CT. Patients with negative PET-2 (score 1–3) were randomly assigned to continue ABVD or omit bleomycin (AVD) in cycles 3–6. Those with a positive PET-2 (score 4–5) received six cycles of BEACOPP-14 or four cycles of BEACOPP_esc_. The 3-year PFS rate (85.7% (95% CI 82.1–88.6%) vs. 84.4% (95% CI 80.7–87.5%)) and 3-year OS rate (97.2% (95% CI, 95.1–98.4%) vs. 97.6% (95% CI, 95.6 to 98.7%)) were similar in the ABVD and AVD groups. The 172 patients with positive findings on the interim scan received BEACOPP as previously mentioned. With this treatment, 74.4% of patients had negative findings on a third [^18^F]FDG-PET/CT. At the end of the study, 3-year PFS was 67.5% and 3-year OS was 87.8%. These results were very similar to those seen in the US Intergroup Trial, S0816, which used a similar approach for patients with stage III/IV disease [27]. A total of 358 HIV-negative patients were enrolled. A PET scan was performed after the first two cycles of ABVD. PET-2-negative patients (score 1–3) received an additional four cycles of ABVD, whereas PET-2-positive patients (score 4–5) were switched to six cycles of BEACOPP_esc_. The 64% 2-year estimate for PFS for the PET-2-positive patients was far superior to the 15–30% 2-year PFS for patients continuing ABVD reported in the literature, and also surpassed the 48% 2-year PFS threshold set as a goal in the trial. These different trials are summarized in Table 3.

In a phase III trial (GITIL/FIL HD 0607) including 782 patients [28], Gallamini and colleagues studied a risk-adapted strategy after two cycles of ABVD. In the case of positive PET-2, patients (*n* = 150) were randomly assigned to four cycles of BEACOPP_esc_ followed by four cycles of BEACOPP_std_ with or without rituximab. ABVD was continued in patients with a negative PET-2 (*n* = 630). The 3-year PFS for all patients, those with a positive PET-2, and those with a negative PET-2 was 82%, 60%, and 87%, respectively (*p* < 0.001). The 3-year PFS for patients assigned to BEACOPP (with a positive PET-2) with or without rituximab was 63% versus 57%, respectively (*p* = 0.53).

In the HD0801 Trial [29], patients with an interim FDG-PET score of three or more proceeded to undergo ifosfamide-based salvage therapy followed by autologous or allogeneic stem cell transplantation. The percentage of patients showing a positive PET-2 (20%) was similar to that reported in previous studies. In the cohort of PET-2-negative patients, the 2-year PFS was 81%, whereas in PET-2-positive patients, the 2-year PFS increased from 12% of the historical control to 74% (76% on an intention-to-treat analysis). Note that the inclusion of patients with a score of three in the group for escalation may have influenced these results.

After all these different studies using therapeutic escalation and de-escalation strategies, the choice of first-line treatment in patients with advanced cHL remains controversial as there are no markers or upfront risk factors to guide treatment choices, since low-risk patients also benefit from intensified treatment.

## 4. New Drugs

In order to further improve front-line treatments and reduce toxicity, new drugs have been studied. These new drugs include brentuximab vedotin and checkpoint inhibitors such as nivolumab or pembrolizumab. These have been combined with ABVD or BEACOPP in first-line treatment.

### 4.1. Brentuximab Vedotin

Brentuximab vedotin is an antibody–drug conjugate that delivers an antineoplastic agent that results in apoptotic cell death selectively in CD30-expressing tumor cells. The randomized ECHELON-1 Study compared ABVD and brentuximab vedotin plus doxorubicin, vinblastine, and dacarbazine (A + AVD) in patients with previously untreated stage III or IV cHL [30]. The median follow-up was approximately 25 months and showed a significantly higher PFS in the A + AVD group than in the ABVD group (2-year modified PFS rate: 82.1% (95% CI 78.8–85.0%) vs. 77.2% (95% CI 73.7–80.4%), respectively). This improvement in PFS was seen at the price of increased adverse events such as neutropenia (58% vs. 45%) and febrile neutropenia (19% vs. 8%) in the A + AVD group compared to the ABVD group. Twenty percent of patients in the A + AVD group presented grade 2 peripheral neuropathy and 11% presented grade ≥ 3 peripheral neuropathy. In this trial, the 4.9% improvement in tumor control came at the cost of more toxicity, including polyneuropathy and neutropenia, necessitating the administration of granulocyte colony-stimulating factor. An update of the ECHELON-1 Study was recently published, which showed a 3-year PFS rate of 83.1% with A + AVD and 76.0% with ABVD (*p* = 0.005) (Figure 2) [31]. A significant improvement in PFS was also observed for patient subgroups aged < 60 years or PET-2-negative.

The GHSG also tested brentuximab vedotin in combination with a modified BEACOPP_esc_ regimen in first-line advanced cHL with the aim of reducing toxicity. In this randomized phase 2 study [32], two front-line treatments were studied: brentuximab vedotin plus etoposide, doxorubicin, cyclophosphamide, procarbazine, and prednisone (BrECAPP) and brentuximab vedotin plus etoposide, doxorubicin, cyclophosphamide, dacarbazine, and dexamethasone (BrECADD). Patients with newly diagnosed advanced cHL between 18 and 60 years of age were randomly assigned to each group. In the BrECAPP group, 86% of patients achieved CR after chemotherapy and 94% had CR as their final treatment outcome. In the BrECADD group, 88% of patients achieved CR after chemotherapy and as their final treatment outcome. The BrECADD regimen was associated with a more favorable toxicity profile and was, therefore, selected to challenge standard BEACOPP_esc_ for the treatment of advanced cHL in the phase 3 HD21 Study by the GHSG, which aims to further reduce treatment-related morbidity.

### 4.2. Checkpoint Inhibitors

Nivolumab is a human immunoglobulin G4 monoclonal antibody that binds to the PD-1 receptor and blocks its interaction with PD-L1 and PD-L2, resulting in decreased tumor growth by inducing PD-1 pathway-mediated inhibition of the immune response. In Cohort D of the CheckMate 205 Study [33], patients with previously untreated cHL first received nivolumab monotherapy and then nivolumab plus AVD (N-AVD) combination therapy. The ORR at the end of treatment was 84% (95% CI 71–93%), with 67% (95% CI 52–79%) achieving CR. At the end of monotherapy and after two combination cycles, the ORR was 69% and 90%, respectively. The most common grade 3–4 adverse effects were neutropenia (49%) and febrile neutropenia (10%). The study confirmed that nivolumab monotherapy followed by N-AVD had a safety profile with no new safety signals. Based on these results, a phase 3 randomized trial is currently being conducted in the US that compares six cycles of brentuximab vedotin plus AVD versus six cycles of N-AVD, with the aim to recruit 987 patients (clinicaltrials.gov (accessed on 7 July 2021) NCT03907488).

Pembrolizumab is a humanized anti-PD-1 monoclonal antibody that has been investigated in cHL. KEYNOTE-087 investigated the efficacy and safety of pembrolizumab in patients with relapsed/refractory cHL in a phase 2 study [34]. Patients had a median age of 35 years and had received a median of four previous lines of therapy. The ORR was 69.0% (95% CI 62.3–75.2%) and the CR rate was 22.4% (95% CI 16.9–28.6%). Given the good results of the treatment in refractory patients, a multicenter, single-arm, phase 2 trial was conducted with pembrolizumab monotherapy followed by AVD in previously untreated patients > 18 years of age with early unfavorable and advanced stage cHL [35]. The main endpoints were to determine the CMR rate and safety. Twelve patients with early unfavorable disease and 18 patients with advanced stage disease were treated with three cycles of pembrolizumab monotherapy followed by AVD for 4–6 cycles. The CMR rate was 37% after pembrolizumab monotherapy and 100% after two cycles of AVD. All patients maintained their response at the end of treatment. No patients received consolidation radiotherapy. The immune-related adverse events were grade 1 rash (*n* = 6) and grade 2 infusion reactions (*n* = 4). Thus, pembrolizumab monotherapy followed by AVD was effective and safe in patients with newly diagnosed cHL, even in those with bulky disease.

## 5. Conclusions

First-line treatment of advanced cHL remains a major field for discussion. The use of PET-adapted strategies has allowed a reduction in the toxicity of both the ABVD and BEACOPP_esc_ regimens. Nevertheless, a longer follow-up of the different studies would be useful to explore the final impact of BEACOPP_esc_ and ABVD on fertility, second cancers, and OS. Currently, the use of front-line strategies varies between countries. Given the relevant acute toxicity of BEACOPP_esc_, appropriate surveillance and supportive care must be available when this protocol is used. As mentioned earlier, in young people, consideration must also be given to medium- and long-term toxicity, which can affect fertility, and the development of new cancers. The patient must be warned of the choice between these different regimens and the consequences that result from them. Many studies evaluating conjugated antibodies and checkpoint inhibitors are underway. There are promising results with brentuximab vedotin, nivolumab, and pembrolizumab. However, the cost associated with the use of these new drugs remains a major concern in many countries, and the incorporation of these new drugs into treatment algorithms still remains limited.

## Figures and Tables

**Figure 1 cancers-13-03745-f001:**
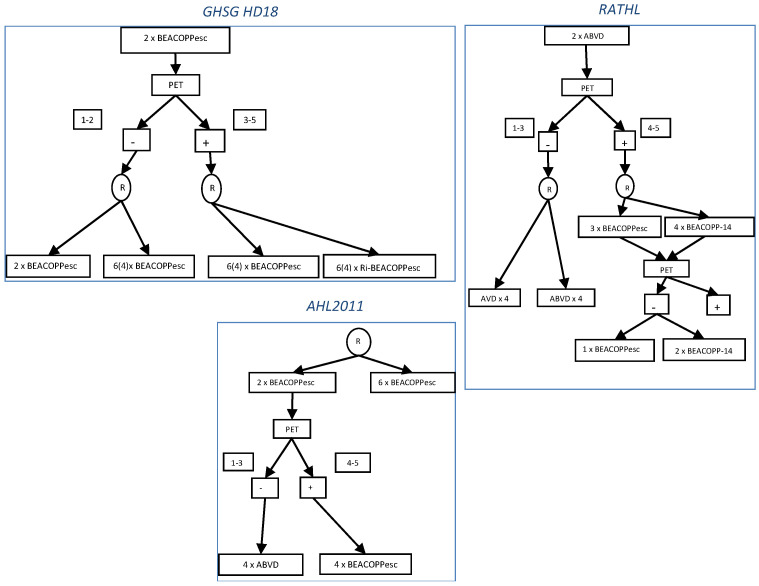
De-escalation and escalation strategy in the GHSG HD18, RATHL, and AHL2011 Trials. ABVD = doxorubicin, bleomycin, vinblastine, and dacarbazine; BEACOPP = bleomycin, etoposide, doxorubicin, cyclophosphamide, vincristine, procarbazine, and prednisone; BEACOPPesc = escalated BEACOPP; BEACOPP-14 = BEACOPP every 14 days; GHSG = German Hodgkin Study Group; PET = positron emission tomography; R = randomization; Ri = rituximab.

**Figure 2 cancers-13-03745-f002:**
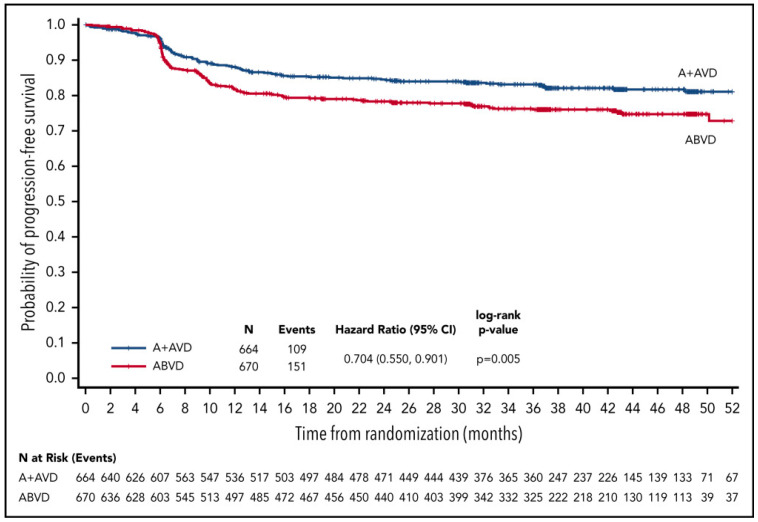
Kaplan–Meier curves of PFS for intent-to-treat patients receiving A + AVD or ABVD in the ECHELON-1 Study. Hazard ratio is for 3-year PFS comparison between groups (reproduced with permission after [31]). A + AVD = brentuximab vedotin plus doxorubicin, vinblastine, and dacarbazine; ABVD = doxorubicin, bleomycin, vinblastine, and dacarbazine; CI = confidence interval; PFS = progression-free survival.

**Table 1 cancers-13-03745-t001:** EORTC/LYSA and GHSG HL risk group classifications.

	EORTC/LYSA	GHSG
**Risk factors**	Mediastinum-to-thorax ratio ≥ 0.35	Mediastinal mass larger than 1/3 of the maximum thoracic width (A)
	Age ≥ 50 years	Extranodal disease (B)
	ESR > 50 mm/h without B symptoms or >30 mm/h with B symptoms	ESR > 50 mm/h without B symptoms or >30 mm/h with B symptoms (C)
	Involvement of ≥4 out of 5 supradiaphragmatic nodal areas	Involvement of ≥3 out of 11 nodal areas on both sides of the diaphragm (D)
**Treatment group**		
**Earlystage**	I–II without risk factors	I–II without risk factors
**Intermediate stage**	I–II ≥ 1 risk factors	I–IIA with ≥1 risk factors. IIB with risk factors C and/or D, but not A/B
**Advanced stage**	III–IV	IIB with risk factors A and/or B, III/IV

EORTC = European Organization for Research and Treatment of Cancer; ESR = erythrocyte sedimentation rate; GHSG = German Hodgkin Study Group; LYSA = Lymphoma Study Association.

**Table 2 cancers-13-03745-t002:** Summary of studies comparing ABVD and BEACOPP regimens.

Trial	Regimen	No. of Pts	PFS (%)	OS (%)	No. of Second Cancers
**EORTC 20012 Intergroup Trial [7]**	ABVD × 8	275	73 (4y)	87 (4y)	14 (5.1%)
	BEACOPP_esc_ × 4, then BEACOPP_std_ × 4	274	83 (4y)	90 (4y)	25 (9.3%)
**HD2000 Trial [8,9]**	ABVD × 6	99	69 (10y)	85 (10y)	1 (1.0%)
	BEACOPP_esc_ × 4, then BEACOPP_std_ × 2	98	75 (10y)	84 (10y)	6 (6.7%)
	CEC × 6	98	76 (10y)	86 (10y)	6 (6.7%)
**LYSA H34 Trial [10]**	ABVD × 8	80	75 (5y)	92 (5y)	7 (9.1%)
	BEACOPP_esc_ × 4, then ≥ BEACOPP × 4	70	93 (5y)	99 (5y)	2 (2.9%)
**Viviani et al. [11]**	ABVD × 6–8, then reinduction and HDT/ASCT if less than CR or PD	168	73 (7y)	84 (7y)	4 (2.4%)
	BEACOPP_esc_ × 4, then BEACOPP × 4, then reinduction and HDT/ASCT if less than CR or PD	163	85 (7y)	89 (7y)	3 (1.8%)

ABVD = doxorubicin, bleomycin, vinblastine, and dacarbazine; ASCT = allogeneic stem cell transplantation; BEACOPP = bleomycin, etoposide, doxorubicin, cyclophosphamide, vincristine, procarbazine, and prednisone; BEACOPP_esc_ = escalated BEACOPP; BEACOPP_std_ = standard BEACOPP; CEC = cyclophosphamide, lomustine, vindesine, melphalan, prednisone, epidoxorubicin, vincristine, procarbazine, vinblastine, and bleomycin; CR = complete response; EORTC = European Organization for Research and Treatment of Cancer; HDT = high-dose therapy; LYSA = Lymphoma Study Association; OS = overall survival; PD = progressive disease; PFS = progression-free survival; y = years.

**Table 3 cancers-13-03745-t003:** PFS and OS in the main PET-adapted trials.

Trial	Regimen	No. of Pts	Stage III/IV (%)	PFS (%)	OS (%)
**GHSG HD18 Trial [24]**	BEACOPP_esc_ × 2, if PET-2+, randomization to BEACOPP_esc_ × 4–6	219	78	91 (3y)	97 (3y)
	BEACOPP_esc_ ×2, if PET-2+ randomization to BEACOPP_esc_ + rituximab × 4–6	220	75	93 (3y)	94 (3y)
**LYSA AHL2011 Trial [25]**	BEACOPP_esc_ × 2, if PET-2–, ABVD × 4	319	88	88 (2y)	96 (5y)
	BEACOPP_esc_ × 2, if PET+, BEACOPP_esc_ × 4	49			
	BEACOPP_esc_ × 6 (no PET adaptation)	401		92 (2y)	95 (5y)
**RATHL Trial [26]**	ABVD × 2, if PET-2–, randomization to AVD × 4	470	59	86 (3y)	97 (3y)
	ABVD × 2 , if PET-2–, randomization to ABVD × 4	465	58	84 (3y)	98 (3y)
	ABVD × 2, if PET-2+, BEACOPP_esc_ × 4 or BEACOPP-14 × 6	172	58	66 (3y)	88 (3y)
**US Intergroup SWOG Trial S0816 [27]**	ABVD × 2, if PET-2–, for ABVD × 4	370	100	82 (2y)	NA
	ABVD × 2, if PET-2+, for BEACOPP_esc_ × 6	55		64 (2y)	NA

ABVD = doxorubicin, bleomycin, vinblastine, and dacarbazine; AVD = doxorubicin, vinblastine, and dacarbazine; BEACOPP = bleomycin, etoposide, doxorubicin, cyclophosphamide, vincristine, procarbazine, and prednisone; BEACOPP_esc_ = escalated BEACOPP; BEACOPP-14 = BEACOPP repeated every 14 days; GHSG = German Hodgkin Study Group; LYSA = Lymphoma Study Association; OS = overall survival; PET = positron emission tomography; PFS = progression-free survival; y = years.

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
