# Peer review of "Review of Treatment Options for the Management of Advanced Stage Hodgkin Lymphoma"

_cancers, 2021, doi:10.3390/cancers13153745_

Round 1
Reviewer 1 Report
The authors adressed all concerns.
Author Response
Thank you very much.
Reviewer 2 Report
The Review by Vellemans and André is a nice review of the current common treatment strategies for advanced HL in the adult population. The paper is nicely written and comprehensive. I only have a few comments:
Line 28: there should be a period (.) at the end of "improvement"
Line 142: what does "recovery of correct ovarian function mean"? please rephrase
Line 143: BEACOPP does not have "high potential" for gonadotoxicity. BEACOPP is extremely gonadotoxic and so I would call it as it is.
Figure 1: for the GHSG HD18, the randomization for PET (+) was BEACOPPesc versus R-BEACOPPesc and that is not clear. Furthermore, it should not be "5R x" - overall the figure could be improved for all the trials because sometimes you separate randomization options into different boxes and sometimes you do not, so I would be consistent.
Table 3: for the RATH trial, the regimens are not well aligned with the results
Other comments. Not sure I would say that the treatment of advanced HL is controversial, as it is not about finding what is right or wrong, but rather what is right or wrong for the individual patient. I would prefer you don't discuss it as a competition, but rather just 2 different alternatives that well-informed patients have to choose from in discussion with their oncologists as it pertains to their priorities.
Finally, if the place allows, a couple of sentences on the outcome of pediatric advanced Hodgkin lymphoma patients would make this even more comprehensive, particularly because adolescents and AYA may end up being treated by medical oncologists while better alternatives with better outcomes and less toxicity may be available in pediatric institutions.
Author Response
Please see the attachment

This manuscript is a resubmission of an earlier submission. The following is a list of the peer review reports and author responses from that submission.
Round 1
Reviewer 1 Report
The authors review controversies on the treatment of advanced Hodgkin lymphoma. This is an interesting topic. The organization of the paper is straightforward, starting with the debate on ABVD versus BEACOPP, then discussing the possibilities of PET-guided approaches and finishing with the outlook on incorporating two recent promising therapeutic approaches, brentuximab and checkpoint inhibitors into first-line therapy of advance Hodgkin lymphoma. However, the style of data presentation is often awkward making the paper not only difficult to read but also to capture the main messages.
The tables are not informative. Tables summarizing the details of major studies on the single topics would be more informative.
Some data are presented in detail as data from the first paper on the prognostic impact of PET by Gallamini in 2006, but not mentioning the paper by Hutchings published at the same time in Blood. Important studies are not mentioned as the SWOG S0816 trial (Press et al JCO 2016) or the HD0801 trial (Zinzani et al, JCO 2016).
New treatment approaches for elderly patients with advanced Hodgkin lymphoma, such as such as nivolumab+BV (Lancet 2020) or BV+Benda (JCO 2020) are not discussed.
Extensive editing of language and grammar is needed.
MINOR Comments
The authors refer to “late myelodysplasia/acute myeloid leukemia” (LINES 23-123). We suggest to use the term “therapy-related myeloid neoplasms” according to WHO 2016 classification .
The topic “BEACOPP and fertility” would deserve some more discussion than being only mentioned the conclusion (LINE 319)
Reviewer 2 Report
In this review, the authors presented the presently available treatments and adverse effects for advanced stage Hodgkin lymphoma. This review is well organized and provided valuable information regarding the efficacy and problem of ABVD, standard BEACOPP, and escalated BEACOPP chemotherapy, and also usefulness of PET-adapted strategies.
Minor point to be corrected
The authors should replace "commma" to "point" between the number like 0,5 to 0.5. This type of error is seen in several parts.
Reviewer 3 Report
The paper by Vellemans et André "Controversies in the management of Advanced stage Hodgkin 2 lymphomas" is a concise review on the management of advanced classical HL in adult patients.
My main criticism would be with the title and tenor of the review. I don't think there are "controversies" about the right treatment of adult patients with HL, rather, there are different choices and philosophies, all of which need to be disclosed to the patient to give him/her the right information to make an informed decision.
BEACOPP versus ABVD has been a decades-long debate and either approach has its advantages and disadvantages and so I would see the value of this review more in shedding light on those so that physicians and patients can make informed decisions.
Here some more punctual comments:
- For section 2, would be nice to see a table comparing the different highlighted trials comparing BEACOPP to ABVD
- For section 3, would be nice to see a figure explaining the escalation and de-escalation strategies for PET directed therapies, including the trials that fall within
- Line 168-177: Not sure about the choice of discussing HD15 here as it is not an escalation or de-escalation study - this trial could be discussed earlier (or later), as the role of PET was in RT administration after completion of all chemotherapy
- Line 179: discussion of HD18 here makes sense, as it is a PET adapted regimen
- LInes 200+: In the trials described PET response was defined differently for different trials, would be important to mention that here, particularly, as different Deauville scores have been used for escalation versus de-escalation
- A table comparing the PET adapted trials - strategies and results would be nice
- Line 250: not sure about the value of including the figure of this
particular paper. Instead, I think a table with the trials using targeted therapy would be more informative - Line 287: please describe the mechanism of action of Nivolumab, as you did for brentuximab and pembro
- I think a couple of sentences describing the outcomes fo adolescents and young adults treated on pediatric protocols that remain higher than any of the trials presented here should be food for thought.
